# Experience of Primary Care Physicians in the Aube Department, France, Regarding the COVID-19 Crisis

**DOI:** 10.3390/healthcare10050852

**Published:** 2022-05-05

**Authors:** Nicolas Braun, Clément Cormi, Michel Van Rechem, Jan Chrusciel, Stéphane Sanchez

**Affiliations:** 1Pôle Territorial Santé Publique et Performance des Hôpitaux Champagne Sud, Centre Hospitalier de Troyes, 10000 Troyes, France; nicolasbraun17@gmail.com (N.B.); jan.chrusciel@hcs-sante.fr (J.C.); stephane.sanchez@hcs-sante.fr (S.S.); 2Service des Urgences, Centre Hospitalier de Troyes, 10000 Troyes, France; michel.vanrechem@hcs-sante.fr; 3University Committee of Resources for Research in Health (CURRS), University of Reims, Champagne-Ardenne, CEDEX, 51095 Reims, France

**Keywords:** general practice, COVID-19, health policy, information literacy, information dissemination

## Abstract

Background: General practitioners (GPs) played a decisive role during the COVID-19 epidemic, particularly in the identification and care of patients at home. This study aimed to describe the primary care physicians’ perceptions of the COVID-19 crisis and to guide future decisions regarding measures to prolong, abrogate, or improve upon methods for crisis management. Methods: This is a cross-sectional study based on a 30-item questionnaire aiming to investigate how primary care physicians (GPs) working in the rural Aube Department experienced the COVID-19 crisis. Results: Among the 152 respondents, 60.5% were not satisfied with the level of information from authorities during the crisis. By multivariate analysis, a feeling of having been adequately informed (OR 21.87, 95%CI 4.14–115.53) and a feeling that non-COVID-19-related diseases were adequately managed (OR 6.42, 95%CI 1.07–38.51) were both significantly associated with an overall satisfaction with the management of the crisis. Conclusion: This study about rural primary care physicians in Eastern France highlights some of the weaknesses of the French healthcare system in terms of the provision of primary care during the epidemic. A leading cause of dissatisfaction was that the information relayed by the health authorities about the disease and its management largely overlooked the primary care providers, many of whom had to rely on traditional media to obtain information.

## 1. Introduction

A new coronavirus, named SARS-CoV-2, and its resulting disease, COVID-19, first emerged in China in late 2019 and has been the origin of the worldwide pandemic that is continuing unabatedly. The disease presents a wide range of symptoms (fever, asthenia, cough, and dyspnea), and its severity depends on various individual risk factors, including age, obesity, chronic disease, and immunosuppression [1]. In the early stages of the pandemic, around one in five patients had severe symptoms, particularly dyspnea, while the majority (80%) presented milder forms of the disease, generally not requiring hospitalization [1,2].

The World Health Organization (WHO) first recognized the pandemic status in March 2020. In France, the first cases appeared in early 2020, increasing exponentially, with the greater Eastern region of France being among the hardest hit areas [3]. France, alongside many other countries, was obliged to impose a nationwide lockdown [4]. By 17 December 2021, the WHO estimated the number of COVID-19 cases worldwide to be over 270 million, of which around 8 million were in France [5]. These numbers have continued to rise with the current resurgence of the disease in the winter months of the northern hemisphere. 

To cope with this major crisis for the healthcare system, nationwide measures that had profound effects on our manner of living and working were implemented, including border closures, lockdown, restrictions on population movement, school closures, mandatory mask-wearing etc. The crisis also led to a profound upheaval in healthcare delivery, notably with major changes to primary and ambulatory care [6], by adapting the rhythm of consultations and appointments, reorganizing patient flows through surgeries and offices, implementing strict hygiene measures, and increasingly using telemedicine solutions [7,8,9].

Primary care physicians are the cornerstone of family medicine in the general population and play a key role during an epidemic such as the COVID-19 crisis [10]. During the peak of the epidemic, they were decisive in identifying patients who could be managed at home and those whose COVID-19 infection required hospitalization. More generally, they participated in promoting the respect of hygiene measures (social distancing, handwashing, and mask wearing) as well as the implementation of prevention strategies, including vaccination [11].

After the peak of the first wave had passed, formal and informal feedback from GPs in the Aube Department of Eastern France suggested that there was some dissatisfaction among GPs about how the crisis had been managed by the local health authorities. In this context, we identified a need to describe primary care physicians’ perceptions of the crisis in order to guide future decisions regarding measures to prolong, abrogate, or improve the management of the crisis. Therefore, we designed a study to analyze how GPs in the Aube Department of Eastern France experienced the COVID-19 crisis using an ad hoc questionnaire. 

## 2. Materials and Methods

### 2.1. Design and Aims

In a cross-sectional study, we used a 30-item questionnaire to investigate how primary care physicians (GPs) working in the rural Aube Department experienced the COVID-19 crisis (see Appendix A). The questionnaire was developed by three authors (NB, CC, and SS) with expertise in general (family) medicine, public health, health systems organization, and telemedicine. The questionnaire was developed based on a review of the literature and knowledge acquired in their daily practice. The questionnaire was tested on a sample group of GPs for relevance and readability, and no modifications were deemed necessary. The results from the pilot test were not included in the present analysis. 

Six main domains were addressed in the questionnaire, namely (1) the information received from the local and national health authorities about the epidemic and the progress of the epidemic, (2) the transmission of information between the main referral hospital in the Department and GPs, (3) the availability of and need for personal protective equipment (PPE), (4) the psychological impact of the crisis, (5) the organisational changes introduced to face the pandemic conditions, and (6) the use of telemedicine. 

The secondary objectives were, first, to analyze the information that the GPs received, the organization of healthcare delivery, and the psychological impact of the crisis on GPs; then, to identify the protective measures available in the GPs’ offices; and finally, to investigate the GPs’ opinions about the value of digital/computer-based solutions and telemedicine.

### 2.2. Participants

GPs in private practice and with practices in the Aube Department of Eastern France were eligible for participation. Although policies for the management of the COVID-19 crisis were decided at the governmental level for the whole country, there may have been specificities in each individual department. We therefore chose to perform our survey among physicians in a single department to obtain a relatively homogenous sample, in terms of their exposure to the challenges of the pandemic in a given geographical area. The survey was distributed electronically using Google Forms by the Order of Physicians of the department and was accompanied by an information letter explaining the purpose of the study. Each physician was sent a unique participation code to avoid duplicates. Non-responders received a reminder after 2 weeks and a maximum of three reminders.

### 2.3. Statistical Analysis

The numerical variables are described as the mean and the standard deviation, and categorical variables as a number and a percentage. Physicians were considered “dissatisfied” if they reported a satisfaction score of 5/10 or lower. The variables were compared between groups defined according to the physicians’ level of satisfaction, using the Chi square test for categorical variables or the Student’s *t* test for numeric variables. The results of the bivariate analysis for categorical variables are reported as unadjusted odds ratios (OR) with 95% confidence intervals (CI). A multivariate analysis was performed using logistic regression, modelling the probability of being satisfied with the management of the COVID-19 crisis. The model included all variables with a *p*-value < 0.20 based on a bivariate analysis, except variables that were correlated with other key model variables (the information sent by hospitals, an increase in decompensations of chronic diseases, and use of personal communication solutions). All analyses were performed using SPSS 27.0 (IBM, Chicago, IL, USA). A *p*-value <0.05 was considered statistically significant.

## 3. Results

From among the 300 physicians contacted, 152 completed the survey online between 15 September and 15 November 2020, yielding a response rate of 50.6%. Among the respondents, 60 (39.5%) declared that they were satisfied and 92 (60.5%) declared that they were dissatisfied with the level of information from authorities during the COVID-19 crisis.

The characteristics of the participants and the results of the survey are detailed in Table 1. None of the socio-demographic variables (age, sex, specialty, or the duration of the practice in the region) were significantly associated with the overall satisfaction score. 

The average satisfaction score was 4/10. Regarding the answers, 56.4% of physicians felt that overall, they were poorly informed about the crisis by the regional health authority (theme 1). This lack of information and the difficulty of accessing reliable information was significantly related to the overall satisfaction score (OR 7.33, 95% CI 2.97–18.12).

Regarding the relations with healthcare establishments (theme 2), three-quarters of participants found that the organization was poor in this respect. When respondents found the information provided by private clinics to be insufficient, they had a significantly lower overall level of satisfaction (*p* < 0.001), whereas the satisfaction with the information provided by the public hospital did not affect the level of overall satisfaction with the management of pandemic (*p* = 0.067). 

In terms of personal protection (theme 3), 66.2% of physicians reported that they had no stock of personal protective equipment (PPE) prior to the beginning of the pandemic (e.g., masks, gowns, and/or hydroalcoholic gel). Furthermore, there was a significant relation between dissatisfaction with the distribution of state-supplied PPE and the overall level of satisfaction (OR 2.66, 95%CI 1.34–5.23).

In terms of mental health (theme 4), 71.1% of respondents declared that the pandemic had caused them mental stress, and this factor was significantly associated with overall level of satisfaction. In addition, almost half of the respondents reported that this stress impacted their management of patients during the crisis.

For the organization of healthcare delivery (theme 5), 33.6% of physicians said they practiced in a health center and 79.2% reported that their organization was adapted to the crisis. Nevertheless, 88.2% reported that they had to change the organization of healthcare delivery in their practice, for example, by setting consultation times according to the motive for consultation (65.8%), postponing non-urgent consultations (46.1%), or postponing follow-up visits for patients with chronic diseases (34.9%). More than half of the physicians reported that they had observed an increase in the number of decompensation episodes among patients with chronic diseases since the first wave.

A total of 83.5% reported that they felt that the management of conditions unrelated to COVID-19 was insufficient, and this was significantly associated with the overall level of satisfaction (OR 3.54, 95%CI 1.24–10.07).

Finally, regarding the use of telemedicine (theme 6), more than half of the respondents reported that they used telemedicine during the pandemic, and for 49.4%, it represented more than 50% of their COVID-19-related activity.

The results of the multivariate analysis by logistic regression modelling of the probability of overall satisfaction with the COVID-19 crisis are presented in Table 2. Through multivariate analysis, a feeling of having been adequately informed (OR 21.87, 95%CI 4.14–115.53) and a feeling that non-COVID-19-related diseases were adequately managed (OR 6.42, 95%CI 1.07–38.51) were both significantly associated with an overall satisfaction with the management of the crisis (Table 2). Conversely, respondents who had mental stress caused by the pandemic were significantly less likely to report that they were satisfied overall (OR 0.14 95%CI 0.03–0.72).

## 4. Discussion

The difficulties encountered when obtaining information about the COVID-19 pandemic were found in this study to be a major contributor to an overall dissatisfaction with the management of the crisis among healthcare professionals working in primary care in the Aube Department in France. These findings are in line with the results of the “Infodemic” survey performed in Belgium in 2020 [12]. In that study, 87% of physicians felt that discussions with other healthcare professionals represented their primary source of information, while the majority of the general population obtained their information principally from traditional media. In September 2020, the WHO invited its member states to develop and implement plans to counter the “Infodemic”, notably by promoting the rapid dissemination of accurate information, based on sound and factual scientific data, to all population groups, especially those at high risk, and by preventing and combatting the propagation of false or misleading information [13]. 

Our study shows that the mental stress generated by the crisis was associated with an overall feeling of dissatisfaction regarding the management of the COVID-19 pandemic. This is in line with the findings of several other studies investigating the mental health repercussions of the COVID-19 pandemic in both healthcare workers and the general public [14,15,16,17,18]. Having sufficient supplies of PPE, as well as accurate and up-to-date information were two factors that are essential pre-requisites to a feeling of security and protection. However, according to the respondents in our study, both these things were lacking. In addition, the fear of giving the virus to family or patients has also been shown to have been a source of stress for healthcare professionals [19,20,21]. In this regard, the use of telemedicine progressed substantially, notably during the periods of lockdown, and represents a viable alternative to traditional face-to-face consultations [9,22]. 

Stress appears to be an unavoidable dimension of the current crisis. Mental stress is known to affect decision-making capacity [23,24]. Ortega-Galan et al. reported a direct link between professional quality of life and perceived stress during the COVID-19 crisis in a sample of professionals from both the hospital and primary care settings, while periods of acute stress can impact patient management [25]. Therefore, for healthcare professionals, recognizing, acknowledging, and quantifying their own level of stress is essential during a time of crisis.

In our study, more than 75% of the respondents indicated that the coordination between community-based medical practitioners and the hospital was insufficient during the pandemic. A report from the French Institute for Research and Information in Health Economics (IRDES) investigating the role of primary care during the COVID-19 epidemic found that existing local structures and ad hoc groups that formed to facilitate coordination and cooperation, which were in place long before the start of the epidemic, were the cornerstone of coordination between primary and hospital-based care. In geographical areas where no such networks or groups existed, it was not possible to mobilize primary care teams to achieve region-wide coordination procedures during the critical phase of the epidemic [26]. 

Among the respondents to our study, all practicing in the Aube Department of Eastern France, only 6.6% declared that they were members of a territorial healthcare professionals group, and 33.6% declared that they practiced in a health center, where other health professionals (e.g., nurses and physiotherapists) are also based. In the context of a plan by the regional health authority to improve the fractionated character of healthcare trajectories in our region (called “My health 2022”) [27], developing professional groups or group practices of this type could enhance coordination between primary care providers and promote improved cooperation. In this regard, a territorial healthcare professionals group in the southeast sector of the Aube Department, created in 2018 and bringing together 23 healthcare professionals, was able to rapidly introduce a dedicated healthcare pathway for COVID-19 patients early on during the pandemic, notably using personal communication tools such as WhatsApp and Dropbox to facilitate the dissemination of information to its members. They were able to develop their own protocols and organizational procedures for their multidisciplinary healthcare center, with the grouping of blood sample collection and transport, for example, while maintaining the continuity of care via nurses and pharmacies in their region. Feedback from their experience reported that this forged strong and lasting links between the participating professionals and laid the foundations for a strong and fruitful cooperation with their local hospital in the town of Bar-sur-Seine [28]. Groups of this type could be a solution to help physicians practicing in isolation avoid feelings of loneliness and abandonment in the practice of their profession [29]. Indeed, in the current context, based on the lessons we have learned from the pandemic, the time of individual practice may now be behind us. Grouping healthcare professionals into networks of cooperation appears to be the key solution in primary care moving forward, encompassing shared information, coordination across providers of patients’ healthcare trajectories, and mutual mental health support between providers. 

Our study has some limitations. First, there is a potential selection bias, whereby those with complaints to voice might have been more motivated to participate. Nevertheless, we had a high response rate (50%) among the registered professionals in the area, thus minimizing this potential bias [30]. The stratification of the results according to the time of response (before the first reminder, after the first reminder, after the second reminder, and after the third reminder) might have helped to reduce the potential for selection bias even further but was unfortunately not possible since the time of response was not recorded. 

Second, the primary endpoint used for this study is a self-reported, subjective evaluation of satisfaction and is not a validated or precise instrument. Other unmeasured confounding factors might also have been unaccounted for. 

## 5. Conclusions

This study among primary care physicians in the Aube Department of Eastern France highlights some of the weaknesses of the French healthcare system in terms of the provision of primary care during the epidemic. A leading cause of dissatisfaction is that the information relayed by the health authorities about the disease and its management largely overlooked the primary care providers, many of whom had to rely on traditional media to obtain information. For the future, enhanced collaboration between healthcare providers within a given geographical area seems to be the most promising solution for implementing crisis-control (management and coordination of primary care). This would also have the advantage of meeting the needs of primary care physicians for greater support, both in terms of patient management and mental health support. Indeed, mental stress is an integral part of the crisis. The increased use of telemedicine solutions also appears to be an interesting avenue to pursue. 

## Figures and Tables

**Table 1 healthcare-10-00852-t001:** Characteristics of the participants and the main results of the survey, according to satisfaction.

Characteristics	Not Satisfied	Satisfied	Total	Unadjusted OR (95%CI)	*p*-Value
	*n* = 92 (60.5%)	*n* = 60 (39.5%)	*n* = 152 (100%)		
Male	57 (62.0%)	41 (68.3%)	98 (64.5%)	0.755 (0.379–1.501)	0.422
Female	35 (38%)	19 (31.7%)	54 (35.5%)		
Age < 50 years	39 (42.1%)	20 (33.3%)	59 (38.8%)	1.472 (0.747–2.898)	0.263
General Practitioner	68 (73.9%)	39 (65%)	107 (70.4%)	0.655 (0.324–1.327)	0.239
Practicing < 25 years	33 (35.9%)	20 (33.3%)	53 (34.9%)	1.119 (0.564–2.220)	0.742
**Theme 1: Information about the COVID-19 epidemic**					
Insufficient	44 (75.9%)	12 (30.0%)	56 (57.1%)	**7.333 (2.967–18.127)**	<0.001
Difficult to access	34 (53.1%)	8 (18.6%)	42 (39.3%)	**4.958 (1.993–12.338)**	<0.001
Overall, not well informed	48 (75.0%)	14 (30.4%)	62 (56.4%)	**6.857 (2.945–15.968)**	<0.001
**Theme 2: Interactions with hospitals during the crisis**					
Information sent by hospitals about the organization of healthcare delivery during the crisis was insufficient	72 (91.1%)	35 (79.5%)	107 (87.0%)	2.645 (0.910–7.689)	0.067
Information sent by private clinics about the organization of healthcare delivery during the crisis was insufficient	64 (86.5%)	26 (60.5%)	90 (76.9%)	**4.185 (1.694–10.338)**	0.001
**Theme 3: Protection**					
No stock of PPE before pandemic	62 (68.1%)	38 (63.3%)	100 (66.2%)	1.238 (0.624–2.457)	0.542
Distribution of state-provided PPE was inadequate	49 (53.3%)	18 (30.0%)	67 (44.1%)	**2.659 (1.337–5.288)**	0.005
**Theme 4: Psychological impact**					
Mental stress	73 (79.3%)	35 (58.3%)	108 (71.1%)	**0.364 (0.177–0.749)**	0.005
Impact on management of patients	37 (46.3%)	19 (48.7%)	56 (47.1%)	1.104 (0.513–2.376)	0.800
Aware of mental health support units	52 (56.5%)	35 (58.3%)	87 (57.2%)	1.077 (0.558–2.080)	0.825
**Theme 5: Organization of COVID-19 care in respondent’s practice or medical center**					
Member of a territorial healthcare professionals group	4 (4.4%)	6 (10.0%)	10 (6.6%)	2.417 (0.652–8.956)	0.196
Work in a healthcare/medical center	31 (33.7%)	20 (33.3%)	51 (33.6%)	0.984 (0.494–1.960)	0.963
Organization adapted	25 (78.1%)	17 (81.0%)	11 (20.8%)	1.190 (0.301–4.703)	1.000
Changed organization within the practice	78 (84.8%)	56 (93.3%)	134 (88.2%)	2.513 (0.785–8.040)	0.111
Consultation times according to motive for consulting	59 (64.1%)	41 (68.3%)	100 (65.8%)	1.207 (0.605–2.409)	0.593
Postponement of follow-up for chronic diseases	35 (38.0%)	18 (30.0%)	53 (34.9%)	0.698 (0.349–1.398)	0.309
Postponement of non-urgent consultations	45 (48.9%)	25 (41.7%)	70 (46.1%)	0.746(0.387–1.438)	0.381
Distribution of healthcare activities between public and private healthcare inadequate	43 (76.8%)	19 (67.9%)	62 (73.8%)	1.567 (0.572–4.288)	0.380
Increase in decompensations of chronic diseases	56 (62.9%)	28 (48.3%)	84 (57.1%)	0.550 (0.281–1.076)	0.079
Management of diseases unrelated to COVID-19 was inadequate	63 (90.0%)	28 (71.8%)	91 (83.5%)	**3.536 (1.241–10.073)**	0.014
Stock of masks	67 (74.4%)	48 (82.8%)	115 (77.7%)	1.648 (0.719–3.778)	0.236
**Theme 6: Telemedicine**					
Use telemedicine	54 (59.3%)	35 (58.3%)	89 (58.9%)	0.959 (0.495–1.860)	0.902
>50% of non-COVID-19 by telemedicine	9 (16.7%)	6 (17.1%)	15 (16.9%)	1.034 (0.333–3.214)	0.953
>50% COVID-19-related activity by telemedicine	27 (51.9%)	16 (45.7%)	43 (49.4%)	0.780 (0.330–1.841)	0.570
Use professional healthcare software solutions	23 (25.0%)	25 (41.7%)	48 (31.6%)	**2.143 (1.067–4.303)**	0.031
Use personal communication solutions	31 (33.7%)	12 (20.0%)	43 (28.3%)	0.492 (0.229–1.058)	0.067

**Table 2 healthcare-10-00852-t002:** Factors associated with overall satisfaction by multivariate logistic regression analysis.

Characteristics	OR	95% CI		*p*-Value
		Lower	Upper	
Having been adequately informed during the pandemic	**21.87**	4.14	115.53	<0.001
Information sent by private clinics about the organization of healthcare delivery during the crisis was insufficient	0.43	0.08	2.23	0.32
Distribution of state-provided PPE was inadequate	0.53	0.13	2.16	0.38
The pandemic caused mental stress for me	**0.14**	0.03	0.72	0.02
I believe the management of non-COVID-19-related diseases was insufficient	**6.42**	1.07	38.51	0.04
Use professional healthcare software solutions	0.94	0.23	3.84	0.93

## Data Availability

The data presented in this study are available from the corresponding author upon request. The data are not accessible to the public due to French legislation, which makes the investigators responsible for data processing.

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
