# Peer review of "Experience of Primary Care Physicians in the Aube Department, France, Regarding the COVID-19 Crisis"

_healthcare, 2022, doi:10.3390/healthcare10050852_

Round 1

Reviewer 1 Report

The topic presented in the paper "Experience of Primary Care Physicians in the Aube Department, France, regarding the COVID-19 Crisis" is intriguing, yet the study raises several concerns for me.

1- Selection bias is a key concern in this study, because unsatisfied persons are probably more eager to engage in such surveys. This prejudice should be taken into account by researchers (might stratifying the data by the time respondents filled the survey: before the reminder, after the first reminder, after the second reminder, and after the third reminder).

2- The sample size is relatively small. More information about the Aube Department in France, such as which territories it covers, is required. Do the 300 GPs that took part in the study represent all of France's GPs?

3- On what grounds were the 30 items (questions) chosen. Did they have external reviewers validate the questions?

4- In the introduction, additional detail on these 30 items is needed, as well as more literature to back up the conclusions of the individual items.

5- Some of the text should be rephrased. Authors should be consistent with some phrases, such as the use of the terms "dissatisfaction" and "satisfaction", which can be confusing sometimes. For example:

"Furthermore, there was a significant relation between dissatisfaction with the distribution of state-supplied PPE, and overall level of satisfaction (OR 2.66, 95%CI 1.34-5.23)."

6- The following paragraph is a little hazy: "Recognizing, admitting, and evaluating stress in oneself, as well as in others, is essential during a time of crisis. This awareness is necessary to meet two crucial objectives. Firstly, it enables appropriate management of the patient (lifestyle and/or mental health support), and secondly, it helps to inform future behaviours through processes such as defusing and debriefing [23, 24]."

Best,

Author Response

Dear Reviewer 1, 

Please find enclosed our responses to your comments. 

Regards, 

Reviewer 2 Report

This paper explains an interesting study aimed to describe the perception of primary care physicians about emergency period like Covid-19 crisis.

Comments and tips

At line 35, you described risk factors but it miss a reference. I suggest you to put some references at the end of the sentence.

From lines 55-57 I suggest you to expand the paragraph describing the reasons they were decisive and how many difficulties the physicians found in the covid19 period.

From line 79 to 87 you explained the statistical analysis. It is very short. I suggest you to expand describing in detail each statistical technique used, like mean standard deviation, qualitative analysis, chi square and t test, Multivariate logistic regression analysis....

In result section you included a big table with different data extracted from survey, but they are not explained a lot in the next paragraph. I suggest you to explain each section (Theme 1, Theme 2,......) in order to compare different physicians perceptions.

At page 5 you missed to insert a lable with the descriprion of the table (ex. Table 2.........)

Author Response

Dear Reviewer 2, 

Please find enclosed our responses to your comments. 

Regards, 

Reviewer 3 Report

  1. Duration of the study is important and the author has not mentioned it.
  2. Line number 19: The author has written 95CI 4.14-115.53. Replace it with 95% CI 4.14-115.53
  3. Line number 96: Table 1: The author presented the data of male participants. If so, then mention the data of female participants. Otherwise replace it with total participants because, in line number 94, the author mentioned that the socio-demographic variable (e.g age, sex, etc) has no significant association with the overall satisfaction score.
  4. Line 126-127: Correct the (OR 21.87,95CI14-115.53) as (OR 21.87,95% CI 4.14-115.53).
  5. It will be better if the author represents the data graphically.

Author Response

Dear Reviewer 3, 

Please find enclosed our responses to your comments. 

Regards, 

Round 2

Reviewer 1 Report

Authors have addressed most of the raised issues, thus I vote for publication in the current form.